# Genetic Predisposition to Elevated Levels of Circulating ADAM17 Is Associated with the Risk of Severe COVID-19

**DOI:** 10.3390/ijms242115879

**Published:** 2023-11-01

**Authors:** Mengyu Pan, Isabel Goncalves, Andreas Edsfeldt, Jiangming Sun, Per Swärd

**Affiliations:** 1Cardiovascular Research-Translational Studies, Department of Clinical Sciences, Lund University, Jan Waldenströms Gata 35, 205 02 Malmö, Sweden; mengyu.pan@med.lu.se (M.P.); isabel.goncalves@med.lu.se (I.G.);; 2Department of Cardiology, Skåne University Hospital, 205 02 Malmö, Sweden; 3Wallenberg Center for Molecular Medicine, Lund University, 221 00 Lund, Sweden; 4Clinical and Molecular Osteoporosis Research Unit, Departments of Orthopaedics and Clinical Sciences, Skåne University Hospital, Lund University, 205 02 Malmö, Sweden; per.sward@med.lu.se

**Keywords:** COVID-19 severity, ADAM17, extracellular, genome-wide association study, Mendelian randomization, causality association

## Abstract

High levels of ADAM17 activity have emerged as an important mediator in severe COVID-19. This study aims to characterize eventual causal relationships between ADAM17 and COVID-19. Using Mendelian randomization analyses, we examined the causal effects of circulating ADAM17 on COVID-19 outcomes using summary statistics from large, genome-wide association studies of ADAM17 (up to 35,559 individuals) from the Icelandic Cancer Project and deCODE genetics, as well as critically ill COVID-19 patients (cases: 13,769; controls: 1,072,442), hospitalized COVID-19 patients (cases: 32,519; controls: 2,062,805) and reported SARS-CoV-2 infections (cases: 122,616; controls: 2,475,240) from the COVID-19 Host Genetics Initiative. The Mendelian randomization (MR) analyses demonstrated that a 1 standard deviation increase in genetically determined circulating ADAM17 (extracellular domain) was associated with an increased risk of developing critical ill COVID-19 (odds ratio [OR] = 1.26, 95% confidence interval [CI]:1.03–1.55). The multivariable MR analysis suggested a direct causal role of circulating ADAM17 (extracellular domain) in the risk of developing critical COVID-19 (OR = 1.09; 95% CI:1.01–1.17) when accounting for body mass index. No causal effect for the cytoplasmic domain of ADAM17 on COVID-19 was observed. Our results suggest that an increased genetic susceptibility to elevated levels of circulating ADAM17 (extracellular domain) is associated with a higher risk of suffering from severe COVID-19, strengthening the idea that the timely selective inhibition of ADAM17 could be a potential therapeutic target worthy of investigation.

## 1. Introduction

Infection with severe acute respiratory syndrome coronavirus 2 (SARS-CoV-2) can range from asymptomatic to severe with a risk of acute respiratory distress syndrome (ARDS), multiorgan failure and death [1]. The entrance of SARS-CoV-2 into host cells is facilitated by the binding of the viral S-protein to the active surface domain of angiotensin-converting enzyme 2 (ACE2) [2]. Normally, ACE2 exerts cardiovascular protective effects by cleaving angiotensin (ANG) II into ANG 1-7, which has vasodilatory, anti-inflammatory and anti-fibrotic effects. Various inflammatory stimuli and the SARS-CoV-2 S-protein have been shown to increase A Disintegrin and A Metalloproteinase 17 (ADAM17) activity, leading to the shedding of the catalytically active ectodomain of ACE2 [3,4,5]. *ADAM17* gene expression is normally high in the lung (www.gtxportal.com, (accessed on 1 January 2023)). Increased activity of ADAM17 could, apart from inducing increased ACE2 shedding, play a pivotal role in inducing COVID-19-associated lung inflammation by shedding membrane-bound tumor necrosis factor (TNF) alpha, interleukin (IL) 6R, TNF receptors (TNFRs) and other pro-inflammatory mediators, thereby contributing to the cytokine storm observed in severe COVID-19 [6]. Overactivation of ADAM17 activity may be of particular importance in COVID-19, given the direct activation of this protease by the virus. Also, the inhibition of ADAM17 activity in vitro has been demonstrated to effectively counteract SARS-CoV-2 infection [7]. Collectively, these findings underscore the growing significance of ADAM17 as a pivotal mediator in the context of severe COVID-19 [8,9,10]. However, the affirmation of this statement through population studies remains constrained as there is limited evidence indicating whether ADAM17 levels can serve as prognostic indicators for severe COVID-19 outcomes.

Circulating ADAM17 retains its proteolytic activity and may reflect tissue-bound ADAM17 [10]. Linking high levels of ADAM17 to increased risk of severe COVID-19 would strengthen the perception that ADAM17 inhibition could be a potential therapeutic target to reduce COVID-19 severity. The regulation of membrane-bound ADAM17 is highly complex. At the post-translational level, interactions with native inhibitors, native activators, adapter proteins and phosphorylation status are essential for transport to the cell surface [6]. Of particular importance, iRhom2 upregulation is required for ADAM17 activity under pro-inflammatory conditions [3]. Therefore, to answer the query of ADAM17′s potential role in severe COVID-19 in humans, it is more relevant to assess the levels of membrane-bound ADAM17 rather than the gene expression of *ADAM17*. It is also important to discriminate between the cytoplasmic and extracellular domains, given that the cytoplasmic domain may be hidden from detection in plasma samples [10]. Therefore, by employing Mendelian randomization (MR), we aimed to investigate if there is a causal relationship between circulating ADAM17 (cytoplasmic and membrane-bound domains) and the risk of severe COVID-19, using the latest genome-wide association studies (GWAS) of ADAM17 [11] and COVID-19 [12].

## 2. Results

A total of 35,559 Icelanders were recruited in the GWAS of circulating ADAM17, with 52% of them originating from the Icelandic Cancer Project and 48% from the deCODE study [11]. The participants constitute over 15% of the adult population of Iceland, and the timing of the sample collection, regarding cancer onset, diagnosis and progression, is unlikely to have a major impact on the GWAS of ADAM17 [11,13]. In the present study, there was some overlap between the samples used for the GWAS of ADAM17 and the GWAS of COVID-19, which included cases of hospitalized COVID-19 patients that were either critically or moderately ill and cases from all reported SARS-CoV-2 infections (termed, respectively, critical COVID-19, hospitalized COVID-19 and SARS-CoV-2 reported infections; see Materials and Methods section). The GWAS of hospitalized COVID-19 and SARS-CoV-2 reported infections included subjects (89 cases, 274,322 controls) from the deCODE study. The GWAS of critical COVID-19 did not include subjects from the deCODE study (Appendix A). Hypothetically, the largest possible number of overlapping samples accounted for 48% of total subjects in the GWAS of ADAM17, 0.8% of subjects in the GWAS of hospitalized COVID-19, 0.7% of subjects for the GWAS of SARS-CoV-2 reported infections and 0% of subjects for GWAS of critical COVID-19.

There were six and four variants associated with circulating ADAM17 in the extracellular and the cytoplasmic domain, respectively (linkage disequilibrium [LD] r^2^ < 0.001, *p* < 5 × 10^−8^ for extracellular domain; *p* < 1 × 10^−6^ for cytoplasmic domain). These variants were used as instrument variables to examine the effect of circulating ADAM17 on COVID-19, with an average F-statistic of 90 and 30 for the extracellular and cytoplasmic domains, respectively (Table 1, Appendix A).

Our primary MR analysis (Figure 1a, Appendix A) showed that a 1 SD increase in genetically determined circulating ADAM17 (extracellular domain) was associated with an increased risk of developing critical COVID-19 (odds ratio (OR) = 1.26, 95% CI: 1.03–1.55, inverse-variance weighted (IVW) method). This potential causal effect was also observed in sensitivity MR analyses, such as the weighted median and RAPS, but the confidence intervals became wider when using MR-Egger. This could possibly be due to a lower value of the $I_{GX}^{2}$ statistics, suggesting that a “no measurement error” assumption may be violated. No heterogeneity in the single-nucleotide polymorphism (SNP) effect was found, nor was directional horizontal pleiotropy detected (Appendix A). Circulating ADAM17 (extracellular domain) showed a trend in the risk of being hospitalized for COVID-19 (OR = 1.09, 95% CI: 0.99–1.21, IVW method) but was not associated with SARS-CoV-2 infection. Causal effects for the cytoplasmic domain of ADAM17 on COVID-19 were not observed.

Our multivariable MR analysis suggested a direct causal role of ADAM17 (extracellular domain) on critical COVID-19 (OR = 1.09; 95% CI: 1.01–1.17) after accounting for body mass index (BMI), which is a risk factor suggested to have a causal effect on severe COVID-19 [14,15].

In contrast, we did not observe any causal effect of COVID-19 on circulating ADAM17 levels. However, critical COVID-19 seemed to have a marginal negative effect on circulating ADAM17 levels (extracellular domain, coefficient = −0.01, 95% CI: −0.02–0, IVW method). Hospitalization due to COVID-19 was also identified as having a marginal negative effect on circulating ADAM17 levels (extracellular domain, coefficient = −0.02, 95% CI −0.04–0, IVW method) (Figure 1b, Appendix A). However, these effects were not confirmed in the sensitivity MR analyses.

## 3. Discussion

In the present study, we provide evidence for a causal effect of circulating ADAM17 (extracellular domain) on the risk of severe COVID-19. These findings add further knowledge to recent studies showing that increasing levels of ADAM17 substrates (ACE2, TNFR1 and TNFR2) are associated with adverse clinical outcomes in COVID-19 patients [8] and that inhibition of ADAM17 activity in vitro dose-dependently reduces SARS-CoV-2 infection [7]. Collectively, these findings indicate that reducing ADAM17 activity has the potential to alleviate the severity of critical COVID-19 cases.

Overall, in the case of the extracellular domain of ADAM17, we identified six instrumental variables that are strongly associated with severe COVID-19. These instrumental variables have been reported to be linked to forced vital capacity, serum butyrylcholinesterase activity, peak expiratory flow and rheumatoid arthritis, as well as immune cell counts (http://www.phenoscanner.medschl.cam.ac.uk/ (accessed on 10 October 2023)). In line with our findings, previous studies have shown that patients with chronic respiratory diseases (associated with reduced forced vital capacity and peak expiratory flow) are at an increased risk of suffering from severe COVID-19 [16]. Others have found that lower serum butyrylcholinesterase activity is associated with severe COVID-19 [17]. Patients with rheumatoid arthritis have also been shown to have an increased risk of suffering from severe COVID-19, even though this association may be affected by factors such as immunosuppressive medications and concomitant interstitial lung disease [18]. Collectively, these genetic associations may contribute to the understanding of ADAM17 and its potential implications in relation to severe COVID-19. Nonetheless, further research is needed to confirm these observational findings.

Previous studies showed that circulating levels of the extracellular domain of ADAM17 were higher in individuals with metabolic syndrome, type 2 diabetes and obesity. Circulating levels of ADAM17 have also been shown to correlate positively with cardiovascular risk markers, such as blood pressure, triglycerides, cholesterol and C-reactive protein [11]. Additionally, genetically explained BMI and smoking were causally related to the risk of being hospitalized due to COVID-19 [12]. High levels of circulating GCNT4, RAB14, C1GALT1C1, CD207 and ABO have also been previously suggested to be causally associated with an increased risk of suffering from critical COVID-19, with comparable odds ratios varying from 1.12 to 1.35 (Appendix A) [19].

Pre-existing cardiometabolic dysfunction is associated with endothelial injury, ongoing inflammation and increased ADAM17 activity [10]. Based on the present study, it could be speculated that a greater pre-existing ADAM17 expression, once infected by SARS-CoV-2, leads to increased shedding of ADAM17 substrates, including ACE2 and TNF-alpha. This could lead to exacerbation of the pre-existing endothelial dysfunction, as well as a further dysregulation of the renin–angiotensin system and the immune system in these patients, which could lead to a more severe state of the infection.

Taking advantage of the large amount of data from GWAS and the MR approach, the present study provides causal evidence for ADAM17’s effects on COVID-19 severity. However, there are still some limitations. For example, circulating ADAM17 levels do not necessarily reflect the cell or tissue activity of ADAM17. Although our findings add to the understanding of COVID-19 pathophysiology, more studies are needed to investigate if the expression and activity of membrane-bound ADAM17 in human cells (such as peripheral blood leukocytes and bronchial epithelial cells) are associated with circulating ADAM17 and if ADAM17 activity can predict the outcome of SARS-CoV-2 infection. Of note, other sheddases may shed ACE2 and inflammatory mediators as well. However, under pro-inflammatory stimuli, the sheddase activity of ADAM17 is favored over ADAM10 due to increased iRhom2 expression [3].

We can only speculate regarding the observed differences between the extracellular and cytoplasmic structural domains of ADAM17 and COVID-19. To the best of our knowledge, it is understood that ADAM17 can undergo swift post-translational activation through its transmembrane domain, whereas its cytoplasmic domain does not contribute significantly to this activation process [20,21,22]. Another explanation could be that part of the measured ADAM17 reflects ADAM17 bound to microparticles from endothelial cells, platelets and leukocytes, where the cytoplasmic domain is hidden from detection [7]. Supporting this hypothesis, strong positive correlations between the extracellular domain of ADAM17, platelets and white blood cell counts were identified [11]. We noticed that 274,411 participants (89 COVID-19 cases and 274,322 controls) from the deCODE study were also included in the GWAS of COVID-19. However, the bias in the estimated causal effect caused by sample overlap (if any) is likely negligible, as estimated by the previously published method (https://sb452.shinyapps.io/overlap (accessed on 8 October 2023)) [23]. It should also be noted that MR may not fully rule out potential collider bias or selection bias [24]. For example, the participants in the GWAS of ADAM17 included an enriched number of cancer patients [11] and cancer may be associated with COVID-19 severity [25]. Part of the study sample used for the GWAS of ADAM17 included participants from the Icelandic Cancer Project. This included patients with prevalent and newly diagnosed cancer, their relatives and a control population, which had been randomly selected from the Icelandic National Registry [13]. This could theoretically render a potential bias to the present study. However, the amount of new cancer cases diagnosed ahead of sampling was less than 5% for all cancers [13]. Also, the newly diagnosed subjects only accounted for a small fraction of the total number of subjects included in the study. In addition, strong associations between cancer and the instrumental variants in MR analyses were not observed. Nevertheless, further studies are needed to ascertain the causal effect of ADAM17 in the risk of severe COVID-19.

Although a protective effect of ADAM17 inhibition has been suggested in a mouse model [9], more evidence in humans is needed. The selective inhibition of ADAM17 in humans has so far been challenging. One main reason for this is the diversity attributed to ADAM17 [26]. It plays an essential role in cell function, regulating the activity of a large number of proteins, including cytokines, receptors, growth factors and adhesion molecules [6]. Also, the high similarity between ADAM17 and other metalloproteases has made it challenging to inhibit ADAM17 selectively [26]. Nevertheless, recent studies indicate that inhibition of iRhom2 can inhibit ADAM17 activity while normal physiological functions are retained [26]. Another approach to inhibit ADAM17 includes the novel ADAM17-directed monoclonal antibody MEDI3622 [9,27]. It was shown that treatment with MEDI3622 attenuated inflammation and lung injury in mice infected with SARS-CoV-2 [9]. In the same study, the authors demonstrated that ADAM17 inhibition led to a simultaneous increase in the viral load of the lung [9]. These findings certainly give rise to some concerns and indicate that the timing of ADAM17 inhibition may be crucial. This would be in line with previous findings showing that a mortality benefit from corticosteroids was observed when >7 days had passed from symptom onset to initiation of corticosteroid treatment [28]. Altogether, in the light of previous studies, the findings of the present study indicate a potentially important role of ADAM17 in determining the risk of severe COVID-19 and suggest that the selective inhibition of ADAM17 in humans could be a potential therapeutic approach, but this warrants further investigation.

In conclusion, an increased genetic susceptibility to elevated levels of circulating ADAM17 (extracellular domain) is associated with a higher risk of severe COVID-19. In agreement with the findings from previous studies, the present study suggests that timely selective inhibition of ADAM17 is a potential therapeutic against severe COVID-19.

## 4. Materials and Methods

### 4.1. Data Sources

Summary statistics for the GWAS of ADAM17 were retrieved from a large GWAS on plasma protein levels measured in 35,559 middle-aged European populations (mean age = 55 years old, standard deviation = 17 years, 57% were women) [11]. Plasma protein levels were measured using the SomaScan aptamers (SomaLogic, Operating Co., Inc., Boulder, CO 80301, USA). To measure ADAM17, two different aptamers were used: one which measured the extracellular and one which measured the cytoplasmic domain. Protein levels were rank-inverse normal transformed, adjusting for age, sex and sample age in the GWAS analysis of ADAM17.

GWAS summary statistics for COVID-19 were retrieved from the COVID-19 Host Genetics Initiative (release 7) [12]. As reported, the GWAS analyses of COVID-19 disease were performed using cases defined by the following information: critically ill cases of COVID-19 (cases: 13,769; controls: 1,072,442, termed as critical COVID-19) who required respiratory support in hospital or who died due to the disease; moderate cases of COVID-19 who were hospitalized (cases: 32,519; controls: 2,062,805, termed as hospitalized COVID-19); and all reported cases of SARS-CoV-2 infection (cases: 122,616; controls: 2,475,240, termed as SARS-CoV-2 reported infections). Controls were genetically ancestry-matched samples without a previously known SARS-CoV-2 infection.

Data for the GWAS of BMI (n = 806,834) were retrieved from a GIANT and UK BioBank meta-analysis [29] for the multivariable Mendelian randomization analysis.

In the present study, GWASs restricted to a European population were used in our MR analyses. The genomic positions for SNPs were harmonized to the same strand in human genome build 19. Ambiguous SNPs and SNPs with a non-inferable forward strand were excluded. Palindromic SNPs with a difference in effect allele frequencies greater than 0.2 were also excluded.

Details on publicly available GWAS summary statistics are provided in the Appendix A.

### 4.2. Instrumental Variable Selection

If not specific, independent variants (linkage disequilibrium (LD) r^2^ < 0.001 in a window of 500 kb) were chosen as instruments where the *p*-value cutoff was 5 × 10^−8^ in respective exposure (ADAM17 or COVID-19), similar to previous studies [30,31]. Instrument variants were not associated with outcomes (*p* > 1 × 10^−5^). When the exposure was ADAM17 (cytoplasmic domain), *p* < 1 × 10^−6^ was used as a cutoff to select instruments since only a small number of SNPs were associated with ADAM17 (cytoplasmic domain) levels. The average F-statistic for the selected instrumental variable was reported to represent its degree of exposure.

### 4.3. Primary Mendelian Randomization Analyses

MR was implemented using the inverse-variance weighted (IVW) method to examine the causal effects of circulating ADAM17 (cytoplasmic domain and cytoplasmic domain) on COVID-19 (critical, hospitalized and SARS-CoV-2 infected). Reversely, the causal effects of COVID-19 on circulating ADAM17 were also tested using the IVW method. Proxy variants were not used if variants were not available in the outcome of the GWAS. Heterogeneity in effect size between instrumental variants was assessed by Cochran’s Q statistic. Potential horizontal pleiotropy was examined by MR-Egger [32].

### 4.4. Sensitivity MR Analyses

Using the same settings as the primary MR analysis, we conducted sensitivity analyses using the weighted median method [33] and MR-Egger [34]. The weighted median method assumes that most genetic variants are valid instrumental variables which are robust to outliers but sensitive to addition/removal of SNPs as instrumental variables. MR-Egger can test for directional pleiotropy and estimate causal effects under the InSIDE (Instrument Strength Independent of Direct Effect) assumption, whereas InSIDE is often not plausible, suggesting that MR-Egger may be less efficient [34].

Additional analyses were also performed using robust adjusted profile score (RAPS) [35] by setting a cut-off *p*-value of 0.001, where other settings were the same as the primary MR analysis. MR-RAPS, which can account for weak instrument bias, extreme outliers and pleiotropic effects of SNPs by assuming pleiotropic effects, were normally distributed. MR-RAPS performed well when the assumption was fulfilled but not when violated.

### 4.5. Multivariable MR Analysis

Multivariable MR analysis was implemented on genetic variants that were associated with either BMI or ADAM17 (*p* < 5 × 10^−8^) using the IVW method. Variants were pruned to exclude associated variants with an LD r^2^ greater than 0.001.

MR analyses were implemented using the R (version 4.1.3) package TwoSampleMR (version 0.5.6) [31].

## Figures and Tables

**Figure 1 ijms-24-15879-f001:**
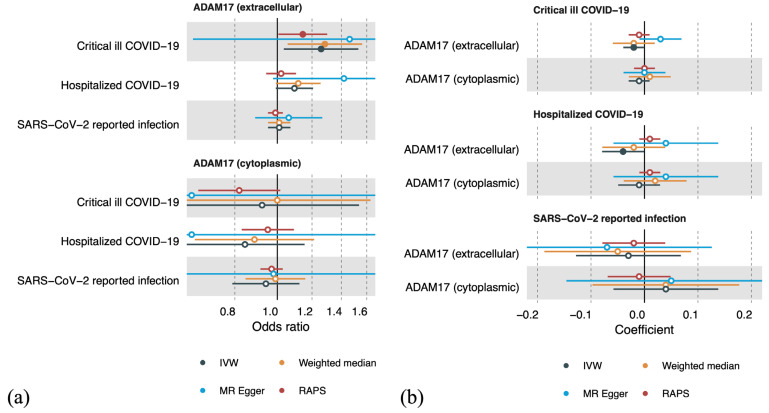
Forest plots showing results from Mendelian randomization analyses. (**a**) Causal effect for circulating ADAM17 on COVID-19. (**b**) Causal effect for COVID-19 on circulating ADAM17. IVW: inverse-variance weighted; MR Egger: Mendelian randomization Egger; RAPS: robust adjusted profile score.

**Table 1 ijms-24-15879-t001:** Characteristics of single-nucleotide polymorphisms used as instrument variables for circulating ADAM17 levels in Mendelian randomization analysis.

Exposure	SNP	Chr	Pos	A1	A2	EAF	Coefficient	SE	P	N	F Statistic
ADAM17 (extracellular)	rs10922098	1	196,664,651	T	C	0.599	−0.129	0.008	6.3 × 10-57	35,338	252.8
	rs7549171	1	197,177,632	G	A	0.242	0.083	0.009	2.9 × 10-19	35,338	80.5
	rs1355538	3	165,505,177	G	A	0.602	−0.066	0.008	3.1 × 10-16	35,338	66.8
	rs6457457	6	31,878,108	T	C	0.065	0.102	0.016	2.6 × 10-10	35,441	39.9
	rs444921	6	31,932,177	T	C	0.139	0.091	0.011	2.3 × 10-15	35,441	62.8
	rs28688825	6	32,587,157	G	A	0.196	0.061	0.010	1.2 × 10-09	35,439	37.0
ADAM17 (cytoplasmic)	rs374896	1	196,692,378	C	T	0.568	−0.040	0.008	3.1 × 10-07	35,339	26.2
	rs17209907	6	32,446,261	T	C	0.386	0.055	0.008	1.5 × 10-11	35,439	45.6
	rs12156434	9	124,133,218	C	T	0.101	−0.066	0.013	5.2 × 10-07	35,374	25.2
	rs55701306	17	16,842,447	T	C	0.048	0.092	0.018	7.3 × 10-07	35,357	24.5

SNP: single-nucleotide polymorphism; Chr: chromosome; Pos: position; A1: effect allele; A2: other allele; EAF: effect allele frequency; SE: standard error; P: *p*-value; N: sample size.

## Data Availability

Summary statistics for the GWAS of circulating ADAM17 are accessible at https://www.decode.com/summarydata/ (accessed on 10 September 2022). Summary statistics for the GWAS of COVID-19 (release 7) are available at https://www.covid19hg.org/results/r7/ (accessed on 15 September 2022).

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
