# Peer review of "Genetic Predisposition to Elevated Levels of Circulating ADAM17 Is Associated with the Risk of Severe COVID-19"

_ijms, 2023, doi:10.3390/ijms242115879_

Round 1

Reviewer 1 Report

Comments and Suggestions for Authors

Dear Authors

Firstly, I commend you on the effort and rigour that has gone into your research. The topic is paramount, especially given the ongoing global efforts to understand and combat COVID-19. Your focus on ADAM17 and its potential association with the severity of the infection has indeed filled a niche in the current scientific landscape.

The review process is an essential facet of advancing our collective scientific understanding, and it is in this spirit that I have approached your manuscript. My primary objective has always been to ensure that the research we put forth to the global community stands up to the highest standards of scientific inquiry. Thus, while my review might contain suggestions and comments for revisions, please understand that they are intended to strengthen your work’s overall impact and clarity.

With the recommended revisions, your manuscript can provide invaluable insights that could contribute significantly to COVID-19 research. I look forward to seeing the revised manuscript and the subsequent contributions your team will make.

Major Comments:

  1. Title and Focus: The title indicates the paper’s central theme, which is the association of circulating ADAM17 with COVID-19 severity. However, it would be beneficial if the authors could further elucidate the clinical and biological significance of ADAM17 in the context of COVID-19 in the introduction.
    • Comment: The title is clear but might benefit from further clarification about the nature or direction of the association.
    • Suggestion: Consider a title like "Increased levels of Circulating ADAM17 are associated with greater severity of COVID-19 symptoms."

2.      Methodology and Cohort: The use of the GWAS study, especially with a large number of Icelanders, adds credence to the paper. Still, it would be beneficial to elaborate on the selection criteria for participants, especially regarding the split between the Icelandic Cancer Project and deCODE.

    • Comment: "GWAS of ADAM17 recruited 35,559 Icelanders where 52% of participants were from the Icelandic Cancer Project and 48% deCODE." There's a lack of detailed selection criteria and potential biases of the cohorts.
    • Suggestion: Delve into the demographic details, comorbidities, and the inclusion-exclusion criteria for participants, which can offer a broader understanding of the study population.

3.      Statistical Analysis: The paper provides detailed statistical analyses, especially in the results section, which is commendable. However, the rationale and validation for selecting specific thresholds (e.g., LD r2<0.001) should be clarified. Additionally, the potential implications of overlapping samples in GWAS studies must be discussed more extensively.

    • Comment: The specific thresholds, like "LD r2<0.001," are not explicitly justified. Also, the statement "However, the bias caused by this (if any) is likely negligible as estimated by the previously published method" requires more detailed justification.
    • Suggestion: Elaborate on the rationale behind these thresholds and how the previously published method validates the negligible bias.

4.      Interpretation of Variants: While the variants associated with ADAM17 are listed comprehensively, a deeper biological interpretation of these variants and their potential role in COVID-19 severity would enhance the paper's depth.

    • Comment: "There were 6 and 4 variants associated with circulating ADAM17 at extracellular and cytoplasmic domain, respectively." While the variants are listed, a more in-depth discussion about their significance would be beneficial, especially in a clinical setting.
    • Suggestion: Perhaps a subsection can be added to discuss the relevance and potential implications of these variants in COVID-19's pathogenesis.

5.      Comparative Analysis: The paper touches upon the causal impact of ADAM17 and its association with other risk factors for severe COVID-19. A more direct comparison, possibly in tabular form, contrasting the risk attributed to ADAM17 against other known factors might offer readers a clearer picture.

    • Comment: "High levels of circulating GCNT4, RAB14, C1GALT1C1, CD207, and ABO were also suggested being causally associated with an increased risk of critical COVID-19." A tabulation contrasting the risks and odds ratios of ADAM17 against these markers can provide clarity.
    • Suggestion: A comparative table detailing the odds ratios, confidence intervals, and the strength of association for each marker will aid in understanding ADAM17's relative importance.
  1. Discussion on Limitations: The paper acknowledges several limitations, notably the fact that circulating ADAM17 might not accurately reflect cellular activity. Exploring the broader implications of these limitations and how they might affect the study's conclusions is essential.
    • Comment: The limitation highlighted in "An important limitation of the study is that circulating ADAM17 does not necessarily reflect the ADAM17 activity in the cells/tissue" is critical. More discussion on the broader implications of this point would be valuable.
    • Suggestion: Discuss how this limitation might impact the study's clinical implications and the potential ways future research can address it.

7.      Potential Biases in the Study: While the authors discuss potential biases like collider bias, the strategies to mitigate or account for these biases during the study's design and analysis should be detailed.

    • Comment: The mention of potential collider bias and selection bias requires more elaboration.
    • Suggestion: Discuss the steps taken during the study design and analysis to account for or mitigate these biases.
  1. Therapeutic Implications: The concluding section touches upon ADAM17's inhibition as a potential therapeutic route. It would be beneficial to provide a more exhaustive discussion or literature review on this topic to substantiate this possible avenue.
    • Comment: "In conjunction with prior research, the timely selective inhibition of ADAM17 emerges as a potential therapeutic avenue..." The therapeutic implications seem significant and thus should be explored in more detail.
    • Suggestion: Dedicate a section discussing the current state of ADAM17 inhibitors, potential side effects, and their applicability in treating severe COVID-19 cases.

Minor Comments:

  1. Table and Figure Consistency:
    • Comment: Ensure that "Table 1" and "Figure 1" are consistently formatted.
    • Suggestion: A unified style guide for tables and figures would enhance the manuscript's overall presentation.
  2. Language and Clarity:
    • Comment: While generally well-written, sentences like "Reversely, causal effects for COVID-19 on ADAM17 were not observed in general" could be made more explicit.
    • Suggestion: Consider rephrasing: “In contrast, we did not observe a general causal effect of COVID-19 on ADAM17."
  3. Referencing:
    • Comment: Some references, are frequently cited, suggesting their importance. These key references should be explicitly discussed.
    • Suggestion: Provide a summary or the main findings of these pivotal references to give readers more context.

Conclusion:

The manuscript by Mengyu Pan et al. offers valuable insights into the role of ADAM17 concerning COVID-19 severity. The approach, using Mendelian randomization, provides a robust framework for investigating causality. However, to strengthen its contributions to the scientific community, the major and minor revisions highlighted above are essential. I recommend a major revision of the manuscript to address the mentioned concerns and areas of enhancement before considering it for publication.

Comments on the Quality of English Language

I have reviewed the manuscript titled "Circulating ADAM17 is associated with COVID-19 severity." While the research topic is compelling and of significant importance, there are some concerns related to the quality and consistency of the English language used throughout the paper. Addressing these concerns will enhance the clarity and readability of your work, ensuring that your research message is conveyed more effectively. Here are my specific comments:

  1. Clarity and Precision: In some sections of the manuscript, sentences could be further refined for clarity and precision. It's vital to ensure that each sentence conveys a clear message, especially in the methods and results sections where precision is crucial.
  2. Consistency in Terminology: There were instances where terminology usage varied. Consistent use of terms is essential to avoid potential confusion for readers. Please ensure that terms, abbreviations, and acronyms are uniformly applied throughout the manuscript.
  3. Grammatical Errors: I observed a few grammatical inconsistencies and errors that, while minor, can distract from the content. It might be beneficial to have the manuscript proofread by a native English speaker or professional editing service.
  4. Sentence Structure: Some sentences could be restructured for better flow and comprehension. Complex sentence structures, especially in the introduction and discussion sections, can make it challenging for readers to grasp the key message.
  5. Use of Tenses: Ensure that the tense usage is consistent throughout. For instance, when discussing previous work, the past tense should be consistently used. When presenting the findings of your study, the present tense would be more appropriate.
  6. Formatting Consistency: While not strictly related to language, ensuring consistent formatting (e.g., spaces after punctuation, consistent use of italics for Latin terms) can enhance the overall presentation of the paper.

In conclusion, I recommend a thorough language and grammar check to improve the overall quality and readability of the manuscript. Addressing the above points will help in presenting your findings in the most effective and clear manner.

Reviewer 2 Report

Comments and Suggestions for Authors

The authors present a relation between COVID-19 and ADAM-17.The conclusion that "the increased circulating ADAM17 at extracellular domain are associated with a high risk of critical COVID-19 strengthening that of ADAM17 may contribute to the risk stratification and a therapeutic option for severe COVID-19" is preliminar, but of interest in the future.

I suggest including some future directions and open questions in this direction.

Comments on the Quality of English Language

No comments

Author Response

Dear Reviewer,

We greatly appreciate your encouragement and advice. In the resubmitted version of the manuscript, we have now included some future directions and open questions in our discussion part.

First of all, we specified our title to “Genetic predisposition to elevated levels of circulating ADAM17 is associated with risk of severe COVID-19 symptoms”. Here we try to give a general idea for the readers that our paper is focusing on “genetic susceptibility” of circulating ADAM17. On the other hand, we can only provide an insight regarding the risk of severe COVID-19 symptoms from a genetic point of view. (seeing page 8, lines176-183). However, the genetic associations are of interest to identify targets for further studies, which could provide evidence for their roles as markers of disease severity (seeing page 7, lines144-166). Additionally, since circulating ADAM17 levels were causally associated with severe COVID-19 symptoms, this suggests that further studies are needed to understand if selective inhibition of ADAM17 could be used as a therapeutic target.  (seeing page 10, lines 215-226).

Reviewer 3 Report

Comments and Suggestions for Authors

Dear authors, many thanks for submitting such type of work. The title and abstract are consistent. The study design is adequate and reproducible. The methodology and the results reported are solid. 

The discussion is well-argued. 

The conclusion supported the results reported. 

The references are consistent. 

Author Response

Dear Reviewer,

We thank the reviewer for the encouraging comments. 

Round 2

Reviewer 1 Report

Comments and Suggestions for Authors

The manuscript can be accepted. 

Comments on the Quality of English Language

The manuscript can be accepted. 

Author Response

Dear reviewer,

We are here to thank you for taking the time to revise our manuscript. Your help is greatly appreciated. Additionally, in this present minor revisions, we have tried our best to polish the language and double-check the grammar. 

Best Regards,

Mengyu, on behalf of all co-authors